# A Dataset of Relighted 3D Interacting Hands

**Gyeongsik Moon**
mks0601@meta.com

**Shunsuke Saito**
shunsukesaito@meta.com

**Weipeng Xu**
xuweipeng@meta.com

**Rohan Joshi**
rohanjoshi@meta.com

**Julia Buffalini**
jbuffalini@meta.com

**Harley Bellan**
harleybellan@meta.com

**Nicholas Rosen**
nicholasrosen@meta.com

**Jesse Richardson**
jesserichardson@meta.com

**Mallorie Mize**
malloriemize@meta.com

**Philippe de Bree**
phillippedebree@meta.com

**Tomas Simon**
tsimon@meta.com

**Bo Peng**
bopeng@meta.com

**Shubham Garg**
ssgarg@meta.com

**Kevyn McPhail**
kmcphail@meta.com

**Takaaki Shiratori**
tshiratori@meta.com

Meta Reality Labs Research

## Abstract

The two-hand interaction is one of the most challenging signals to analyze due to the self-similarity, complicated articulations, and occlusions of hands. Although several datasets have been proposed for the two-hand interaction analysis, all of them do not achieve 1) diverse and realistic image appearances and 2) diverse and large-scale groundtruth (GT) 3D poses at the same time. In this work, we propose Re:InterHand, a dataset of relighted 3D interacting hands that achieve the two goals. To this end, we employ a state-of-the-art hand relighting network with our accurately tracked two-hand 3D poses. We compare our Re:InterHand with existing 3D interacting hands datasets and show the benefit of it. Our Re:InterHand is available in here.

## 1   Introduction

Humans often make two-hand interactions during daily conversation or when interacting with objects. Self-similarity, complicated articulations, and small sizes of hands make analyzing such two-hand interactions greatly challenging. In particular, when the input of an analyzing system is a single image, the problem becomes much more difficult as in most cases, most of a hand is occluded by the other hand.

One fundamental direction to successfully analyze interacting hands is collecting large-scale 3D interacting hands datasets, which contain in-the-wild images and corresponding 3D groundtruth (GT). Unfortunately, this is not trivial. Due to the inherent scale and depth ambiguity, true 3D data is not obtainable from a single 2D observation. In addition, a single 2D observation does not provide enough information of other viewpoints, necessary for the 3D data collection. Therefore, there have been three alternative approaches to collect 3D hand data.

37th Conference on Neural Information Processing Systems (NeurIPS 2023) Track on Datasets and Benchmarks.

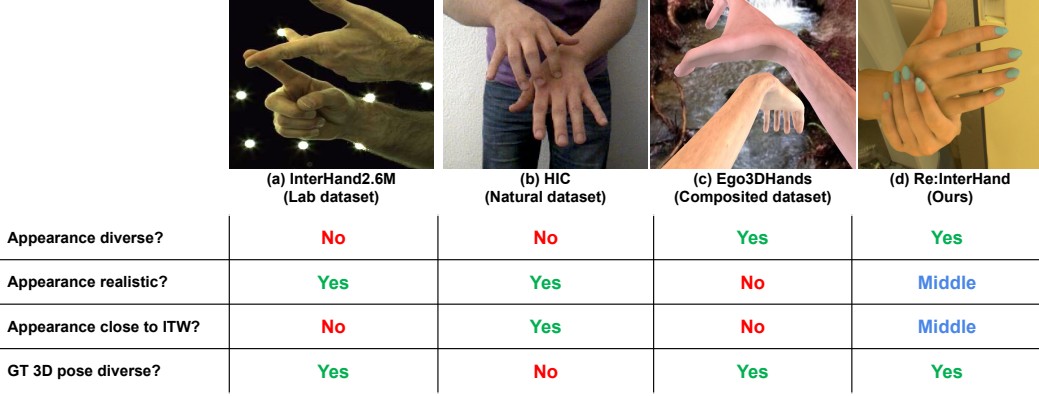

| | (a) InterHand2.6M (Lab dataset) | (b) HIC (Natural dataset) | (c) Ego3DHands (Composited dataset) | (d) Re:InterHand (Ours) |
|---|---|---|---|---|
| Appearance diverse? | No | No | Yes | Yes |
| Appearance realistic? | Yes | Yes | No | Middle |
| Appearance close to ITW? | No | Yes | No | Middle |
| GT 3D pose diverse? | Yes | No | Yes | Yes |

Figure 1: Comparison of datasets from existing data capture approaches ((a), (b), and (c)) and our new Re:InterHand dataset ((d)). ITW represents in-the-wild environments.

## 1.1 Lab datasets

Lab datasets are captured from specially designed studios with hundreds of calibrated and synchronized cameras. InterHand2.6M [30, 29] is the most widely used 3D interacting hands dataset, and it is captured from a studio with 100 calibrated and synchronized cameras. Fig. 1 (a) shows an image example of InterHand2.6M.

**Pros.** They provide large-scale, diverse, and accurate GT 3D poses.

**Cons.** Images have monotonous appearances. The figure shows that images have far and limited diversities of color, backgrounds, and illuminations compared to those of in-the-wild images.

## 1.2 Natural datasets

Natural datasets, such as HIC [52] and RGB2Hands [53], are captured from daily environments with a much smaller number of cameras, for example, a single RGBD camera. Fig. 1 (b) shows an image example of HIC.

**Pros.** As the figure shows, the image appearance is close to those of in-the-wild images.

**Cons.** The diversity and scale of the dataset is limited. Although the capture setup is much lighter than that of lab datasets, bringing the setup and capturing at diverse places is not easy, which makes appearance diversity limited (*e.g.*, in front of desks). As only a few cameras are used, such datasets could not provide accurate annotations for complicated interacting hands. Therefore, they provide simple poses.

## 1.3 Composited datasets

Composited datasets, such as Ego3DHands [23], are a composition of hand images with random background images. The purpose of the composition is to enhance the appearance diversity of lab images or synthesized images. Fig. 1 (c) shows an example of it.

**Pros.** They often have accurate and diverse GT 3D poses as the composition is performed on lab datasets or synthesized datasets.

**Cons.** The figure shows that its image appearances are not realistic due to the light inconsistency between foreground and background.

## 1.4 The proposed *Re:InterHand* dataset

All the existing three approaches have their own limitations. **In this work, we propose Re:InterHand dataset, which complements all three existing dataset collection approaches.** Fig. 1 (d) shows an

Table 1: Comparison of hand datasets that provide GT 3D poses. To count the number of images, we consider images from different viewpoints at the same time step as different ones.

| Datasets | Image appearance | GT | # of subjects | # of images | Two-hand interactions |
|---|---|---|---|---|---|
| Dexter+Object [44] | Natural | 3D fingertip coord. | 1 | 3K | No |
| STB [59] | Natural | 3D joint coord. | 1 | 36K | No |
| EgoDexter [33] | Natural | 3D fingertip coord. | 4 | 3K | No |
| RHD [61] | Composite | 3D joint coord. | 20 | 44K | No |
| PanopticStudio [43] | Lab | 3D joint coord. | N/A | 15K | No |
| FPHA [9] | Natural | 3D joint coord. + 3D obj. | 6 | 105K | No |
| GANerated [31] | Composite | 3D joint coord. | N/A | 264K | No |
| FreiHAND [62] | Natural | MANO | 32 | 134K | No |
| ObMan [15] | Composite | MANO + 3D obj. | 20 | 150K | No |
| EHF [39] | Lab | SMPL-X | 1 | 100 | No |
| HO3D [13] | Natural | 3D joint coord. + 3D obj. | 10 | 78K | No |
| ContactPose [3] | Lab | 3D joint coord. + 3D obj. | 50 | 2.9M | No |
| HUMBI [56, 55] | Lab | MANO | 453 | 24M | No |
| DexYCB [4] | Natural | MANO + 3D obj. | 10 | 582K | No |
| AGORA [38] | Realistic | SMPL-X | 350 | 19K | No |
| DART [8] | Composite | MANO | N/A | 787K | No |
| BlurHand [34] | Lab | MANO | 11 | 156K | No |
| H2O [21] | Natural | MANO + 3D obj. | 4 | 571K | Weak |
| Assembly101 [42] | Natural | 3D joint coord. + action labels | 53 | 111M | Weak |
| AssemblyHands [35] | Natural | 3D joint coord. | 34 | 3M | Weak |
| ARCTIC [7] | Lab | MANO + SMPL-X + 3D obj. | 10 | 2.1M | Weak |
| HIC [52] | Natural | MANO | 1 | 36K | Strong |
| RGB2Hands [53] | Natural | 3D joint coord. wo. fingertips | 2 | 1K | Strong |
| InterHand2.6M [30] | Lab | MANO | 27 | 2.6M | Strong |
| Ego3DHands [23] | Composited | 3D joint coord. + masks | 1 | 55K | Strong |
| **Re:InterHand (Ours)** | Realistic | MANO + masks | 10 | 1.5M | Strong |

image example of our Re:InterHand dataset. Our dataset is constructed by rendering 3D hands with accurately tracked 3D poses and relighting it with diverse environment maps. By using accurately tracked 3D poses from our multi-camera studio, we could secure diverse GT 3D poses. For the relighting, we employ a state-of-the-art hand relighting network [17], which provides diverse and realistic image appearances. The figure shows that our rendered data has close appearances compared to those of in-the-wild images.

## 2 Related works

**3D hand datasets.** Tab. 1 shows comparisons of various 3D hand datasets. Motivated by the Kinect device, early datasets comprise depthmaps [49, 51, 47, 57, 44]. For more practical applications without requiring depth cameras, RGB-based datasets have been introduced. STB [59] includes sequences with simple hand poses. HIC [52] is one of the earliest approaches to address two-hand interactions. RHD [61] consists of synthetically rendered images using commercial software and composited with web-crawled background images. EgoDexter [33] includes sequences with simple hand-object interactions. Panoptic Studio [43] is captured from a specially designed dome, and it contains whole-body humans. FPHA [9] includes hand sequences captured from first-person viewpoints. GANerated [31] is synthetically generated using generative adversarial networks and composited with background images. EHF [39] is a small-scale dataset captured from a multi-camera studio. It includes a whole-body performance of a single subject. ObMan [15] includes simple hand-object interactions. It is synthetically rendered using commercial software and composited with background images. FreiHAND [62] is captured with a portable multi-camera setup in various places. It consists of natural images and composited images. Mueller et al. [32] introduced a synthetic depth map dataset of interacting two hands. YT3D [20] includes web-crawled videos and 3D pseudo-GT of hands. NeuralAnnot [29] introduced 3D pseudo-GT of hands on MSCOCO [24, 18] dataset. Both YT3D and NeuralAnnot fit a 3D hand model [40] to 2D joint coordinates to obtain 3D pseudo-GT.

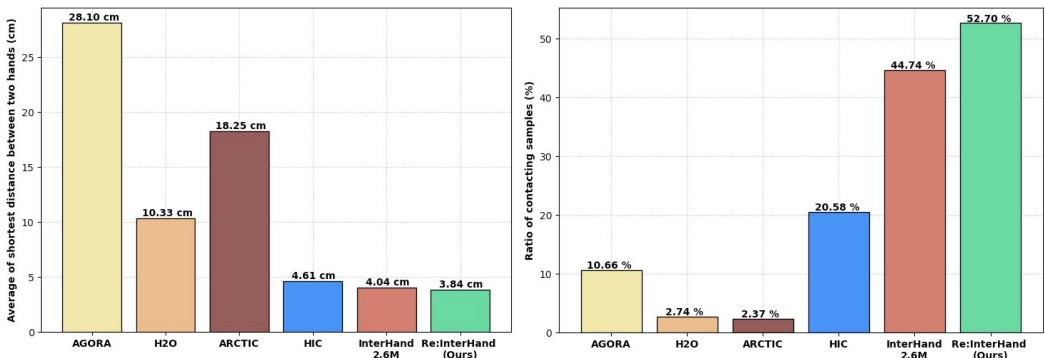

Figure 2: Left: HIC [52], InterHand2.6M [30], and our Re:InterHand have a short distance between two hands. Right: InterHand2.6M [30] and our Re:InterHand have many samples where two hands are in contact. We count a sample as contacting if the shortest distance between the vertices of two hands is smaller than 3 mm. We exclude samples of ARCTIC [7] whose distance between two hands is longer than 1 meter.

They mostly contain single-hand 3D pseudo-GT without 3D relative translation between two hands due to depth and scale ambiguity. HO3D [13] includes 3D hands interacting with various types of objects. RGB2Hand [53] introduced a small-scale 3D interacting hands dataset with 3D joint coordinate annotations without fingertips. InterHand2.6M [30] is a large-scale 3D interacting hands dataset, captured from a specially designed multi-camera studio. ContactPose [3] contains sequences of 3D hands and contact maps, generated from hand-object interactions. HUMBI [56, 55] is a large-scale dataset that provides 3D whole-body annotations, captured from a specially designed multi-camera studio. DexYCB [4] includes large-scale 3D hands interacting with various types of objects.

Ego3DHands [23] is a composition with random background images and rendered two-hand images. H2O [21] contains two hands interacting with objects. AGORA [38] is rendered with 3D scans of people and scenes. Like our Re:InterHand dataset, AGORA considers light consistency between foreground and background, which makes their image appearances realistic. Ego4D [11] includes a huge amount of first-person viewpoint videos; however, it does not provide 3D hand annotations. DART [8] contains rendered images of a single hand with accessories and their texture map, alpha-blended with background images from MSCOCO [24]. Assembly101 [42] contains large-scale videos of 3D hands assembling several objects. AssemblyHands [35] improved Assembly101 [42] with a better annotation pipeline. ARCTIC [7] includes 3D hands and whole-body annotation with 3D objects. BlurHand [34] is made from a subset of InterHand2.6M [30]. It includes blurred hand images and corresponding GT 3D hands.

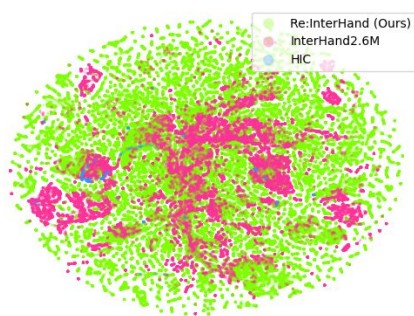

Figure 3: t-SNE of two-hands' 3D pose of our Re:InterHand, Inter-Hand2.6M [30], and HIC [52].

Although there have been many 3D hands datasets introduced, there is a small number of datasets with strong two-hand interactions [52, 53, 30, 23]. Among them, InterHand2.6M [30] and HIC [52] are widely used as RGB2Hands [53] have no 3D fingertip annotations, and images of Ego3DHands [23] are not photorealistic. Some datasets [61, 43, 39, 20, 3, 56, 55, 21, 38, 29, 42, 35, 7, 34] have two-hand annotations; however, they have weak interactions between hands. Fig. 2 shows that only HIC [52], InterHand2.6M [30], and our Re:InterHand have a short distance between two hands and meaningful ratio of contacting samples. Unfortunately, none of such two-hand datasets has achieved the two goals at the same time: 1) rich and realistic image appearances and 2) accurate and diverse GT 3D poses of interacting hands. Our Re:InterHand is the first dataset that achieves the two goals. In addition, Fig. 3 shows that our Re:InterHand has much more diverse 3D interacting hand poses than InterHand2.6M [30], and HIC [52].

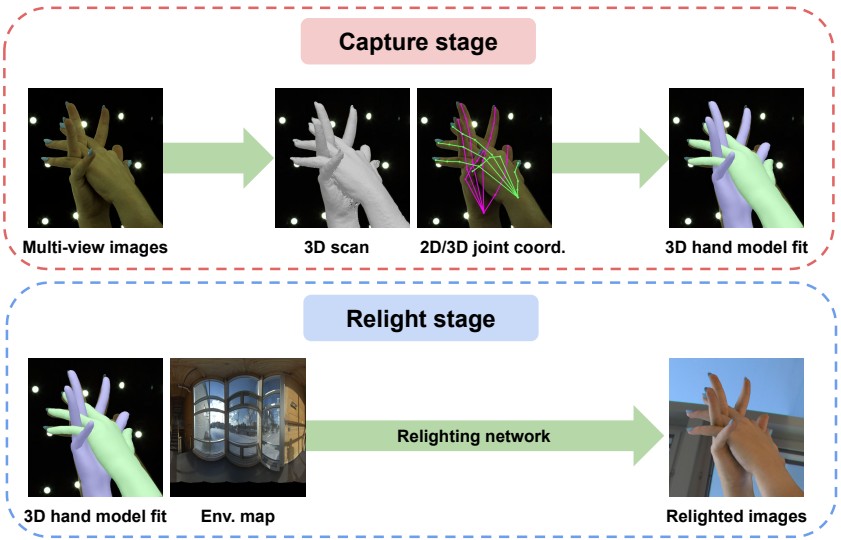

Figure 4: The overall pipeline of constructing our Re:InterHand dataset.

**3D interacting hands recovery.** Due to the absence of large-scale datasets, early works [36, 1, 52, 50, 32, 53] are based on a fitting framework, which fits 3D hand models to geometric observations, such as RGBD sequence [36], hand segmentation map [32], and dense matching map [53]. Inter-Hand2.6M [30, 29] motivated many regression-based methods [41, 58, 22, 14, 5, 6, 19, 25, 26]. Such regression-based methods outperform the above fitting-based approaches while running in real-time. Li et al. [22] introduced a Transformer-based network with the cross-attention between right and left hands. Moon [26] presented a 3D interacting hands recovery network that addresses the domain gap between multi-camera datasets and in-the-wild datasets, which results in robust performance on in-the-wild images.

**Relighting humans.** Several works [54, 46, 60, 60] are proposed to relight faces and bodies; however, these models are not animatable. To enable relighting with animation, Bi et al. [2] presented a deep relightable appearance model for facial avatars. DART [8] provides a dataset of relighted hands; however, their images are not photorealistic as they do not consider light consistency between foreground and background. Iwase et al. [17] introduced an efficient neural relighting system for photorealistic hand relighting using a student-teacher framework and feature-based relighting [37]. We use the relighting system of Iwase et al. [17] due to their high-quality results and rendering efficiency.

## 3 Dataset construction

Fig. 4 shows the overall pipeline for the construction of our dataset. It consists of two stages: *capture* and *relight*.

### 3.1 Capture stage

The capture stage captures hand data from our multi-camera studio. We capture data from 10 subjects, as shown in Fig. 5. Two types of sequences, peak poses and range of motion, are captured following InterHand2.6M [30]. The peak pose is a sequence, which includes a transition from a neutral pose to a pre-defined pose and then transition back to the neutral pose. The purpose of the peak pose is to capture as diverse poses as possible including extreme poses and maximal finger bent. The range of motion is a sequence, which includes natural hand motion driven with minimal instructions, such as waving hands as if friends are coming over. In this way, we could capture both 1) diverse poses from the peak pose sequences and 2) natural hand motion from the range of motion sequences. We provide more image and pose examples of our dataset in the supplementary material.

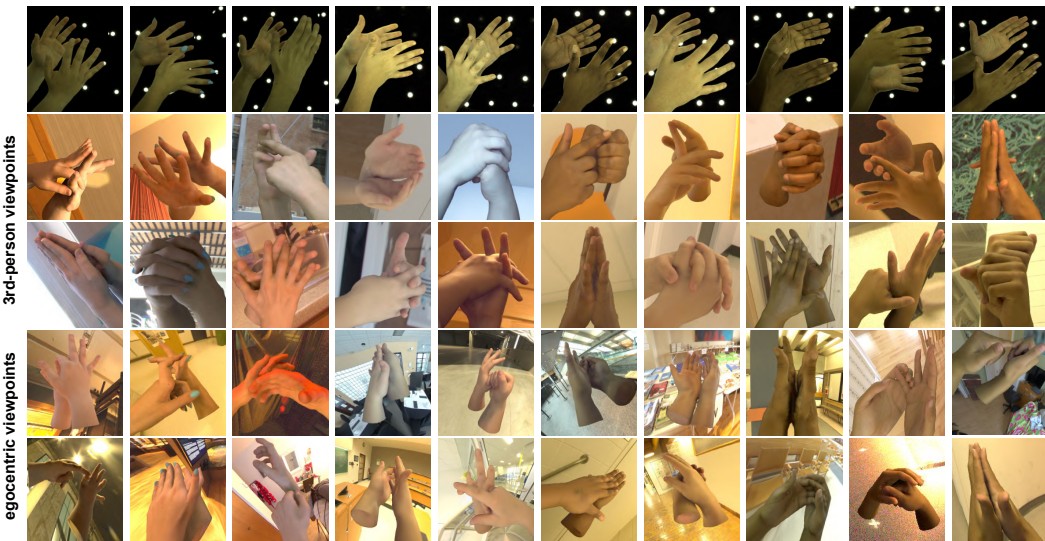

Figure 5: Each column shows images of a subject of our Re:InterHand dataset. For each column, the top image with the neutral pose is from the capture stage, and the remaining images with captured poses are from the relight stage.

**Capture studio.** Our capture studio has 469 lights and 170 calibrated synchronized cameras. All cameras lie on the front, side, and top hemispheres of the hand and are placed at a distance of about one meter from it. Images are captured with $4096 \times 2668$ pixels at 90 frames per second (fps). Following Bi et al. [2], we interleave fully lit frames and partially lit frames at every 3 frames. The capture stage only uses fully lit frames, and the relight stage uses partially lit frames to train the relighting network.

**2D joint coordinates and 3D scans.** We process the raw video data by performing 2D joint detection [45] and 3D scan [12]. The 2D joint detector is trained on our held-out manually annotated dataset, which includes 900K images with rotation center coordinates of hand joints, where our manual annotation tool is similar to that of Moon et al. [30]. Our 2D joint detector has an error of 2.5 pixels in a $1024 \times 667$ image space.

**3D joint coordinates.** InterHand2.6M [30] triangulated detected multi-view 2D joint coordinates with the RANSAC algorithm. We found that their approach suffers from temporally inconsistent results as the triangulation does not take into account the similarity between close frames. For example, some joints could have inconsistent semantic positions across viewpoints due to the failure of the 2D joint detector. In this case, triangulated 3D coordinates of such joints could be very different between close frames if selected viewpoints by RANSAC are different. Instead of triangulation, we train a 3D joint detection network, which takes a voxelized 3D scan of hands and is supervised with multi-view 2D joint coordinates. Our network produces much more temporally consistent and smooth results as inputs of close frames (*i.e.*, voxelized 3D scans) are almost the same.

The network is designed with V2V-PoseNet [27], a state-of-the-art 3D joint detection network from voxelized hands. First, we make two volumes from 3D scans by making 3D bounding boxes around the mean of initially obtained left and right hands' 3D joint coordinates, where the initial ones are obtained with the RANSAC algorithm. Then, we voxelize 3D scans around each 3D bounding box to $(96, 96, 96)$ resolution. The voxelized 3D scans are passed to the V2V-PoseNet, which consists of stacked 3D convolutional layers. We perform soft-argmax [48] to the output of the V2V-PoseNet, which produces 3D joint coordinates in a differentiable way. The obtained 3D joint coordinates are supervised with multi-view 2D joint coordinates by projecting the 3D ones to each viewpoint and calculating $L1$ distance from the 2D ones. We train V2V-PoseNet on all frames, which takes 1 day, and test it on the same frames to obtain 3D joint coordinates of them. Our obtained 3D joint coordinates have an error of 2.0 mm. The errors are measured against our held-out human-annotated set.

**3D hand model fitting.** We additionally obtain 3D meshes of hands as 1) they provide useful surface information that does not exist in the 3D joint coordinates and 2) they are inputs of the relighting network [17]. To this end, we fit 3D hand models, such as MANO [40], to the obtained 3D joint coordinates and 3D scans of the above using NeuralAnnot [29]. The 3D hand model is a parametric model that produces 3D hand meshes from 3D pose and identity (ID) codes. The 3D pose represents 3D joint angles, and ID codes determine 3D hand shape, such as thickness, in the zero pose.

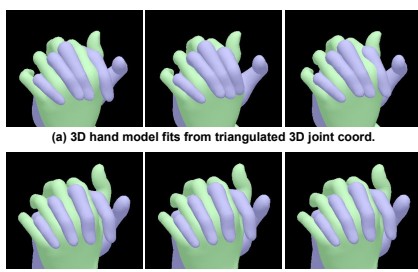

(a) 3D hand model fits from triangulated 3D joint coord.

(b) 3D hand model fits from V2V-PoseNet (Ours)

Figure 6: Comparison of 3D human model fits from (a) triangulation of InterHand2.6M [30] and (b) our V2V-PoseNet. The three frames are consecutive ones, and the time difference between near frames is 0.02 seconds. Given the very short time difference between frames, the three frames should have almost the same 3D hands. (a) not only suffers from the collisions but also suffers from temporal inconsistency between very close frames. On the other hand, (b) does not suffer from the collisions and achieves temporal consistency between close frames.

NeuralAnnot takes a single image and 3D joint coordinates as inputs and outputs 3D pose and ID codes, used to drive 3D hand models. We use the network architecture of Pose2Pose [28] for NeuralAnnot. The 3D pose and ID codes are supervised with the 3D joint coordinates after performing forward kinematics. Also, 3D meshes from the 3D pose and ID codes are supervised with 3D scans by minimizing the closest distance between 3D meshes and 3D scans. Several regularizers, such as 1) $L2$ regularizers to 3D pose and ID codes, which prevents extreme meshes, and 2) a collision avoidance regularizer are applied as well. We separately train NeuralAnnot for each subject, and the ID code is directly optimized, not regressed from the input image and 3D joint coordinates. In this way, 3D hands from the same subject have a consistent ID code. Training NeuralAnnot takes less than 1 hour for each capture. After training NeuralAnnot, we test it on the training set and manually inspect all frames. Frames with wrong fitting results are excluded for the following relight stage. Fig. 6 shows that it produces temporally consistent results. We checked that the MANO meshes from NeuralAnnot have 1.3 mm errors from the 3D scans without any translation/rotation/scale alignments.

### 3.2 Relight stage

After capturing data in the above capture stage, we train a relighting network [17] for each subject following their original training strategy. Following them, we train the relighting networks on single-hand data as 3D hand model fittings are more accurate for the single-hand data than the two-hand data, which makes training the relighting network more stable. Please note that the single-hand data to train the relighting networks are also obtained from NeuralAnnot by training and testing it on single-hand captures. For more details, please refer to Iwase et al. [17]. After training the relighting networks, we use the 3D poses from NeuralAnnot [29] of the above capture stage to render two hands with specified camera parameters. For illuminations, we use 2144 high-resolution environment maps of Gardner et al. [10].

## 4 Dataset release

Our Re:InterHand dataset includes 1) relighted images, 2) non-binary masks, and 3) 3D hand model fittings, as shown in Fig. 7. The relighted images and non-binary foreground masks are from Sec. 3.2, and 3D hand model fittings are from Sec. 3.1. Out of 10 captures, we split 7 captures for the training set and the remaining 3 captures for the testing set.

**Relighted images.** To render relighted images, we first sample cameras out of our 170 cameras for each capture to make overall rendering faster and remove redundancy. To sample cameras, we sum 2D joint detection confidence from Sec. 3.1 for each camera. Then, we pick the top 50 cameras based on the sum of confidence values. In this way, we can exclude cameras where hands are almost not visible. The farthest iterative sampling algorithm samples $N$ cameras from the selected 50 cameras based on the camera positions to obtain as diverse viewpoints as possible. For the frame-based

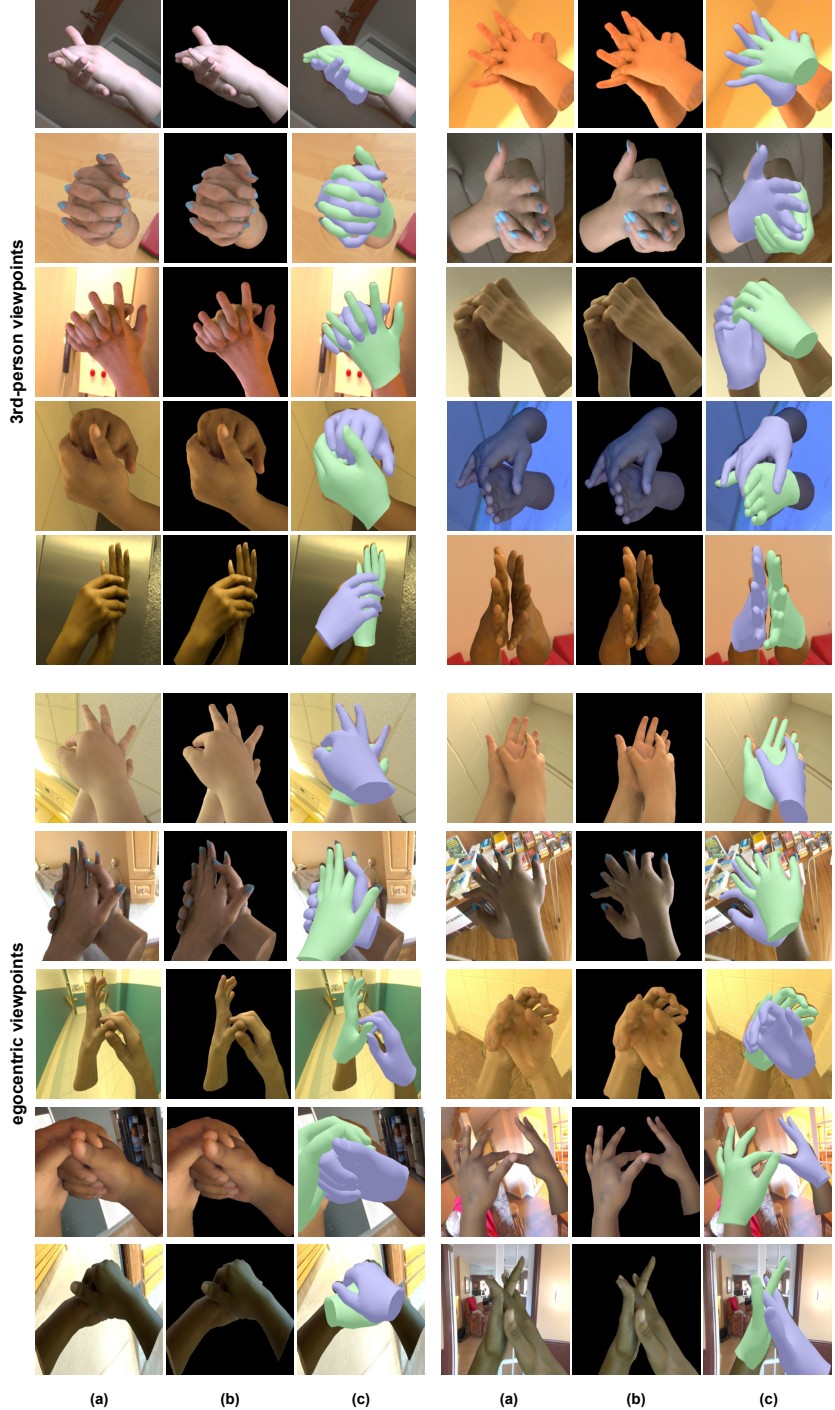

Figure 7: (a) relighted rendered images, (b) masked images, and (c) MANO fitting overlay.

research, we downsample captures at 5 fps and set $N = 20$. Then, we render images with a different environment map for each frame, which results in 493K images. Also, for the video-based research, we set $N = 5$ and render images at 30 fps with a different environment map for each segment, which results in 739K images. For both frame-based and video-based split, images with the same frame index and different viewpoints are rendered from the shared environment map in a multi-view consistent way.

Table 2: RRVE comparison of InterWild [26] trained on different data including variants of our dataset. We use the 3rd-person viewpoint split of our Re:InterHand.

| Training sets | Testing sets | | |
| --- | --- | --- | --- |
| | InterHand2.6M | HIC | Re:InterHand |
| InterHand2.6M + MSCOCO | 19.74 | 23.59 | 37.59 |
| + Capture stage | **19.14** | 23.09 | 34.23 |
| + Capture stage and Composite | 19.50 | 24.37 | 31.30 |
| + Capture stage and Composite (w. AdaIn [16]) | 19.44 | 24.83 | 27.96 |
| **+ Capture stage and Relight stage (Ours)** | 19.40 | **21.36** | **20.07** |

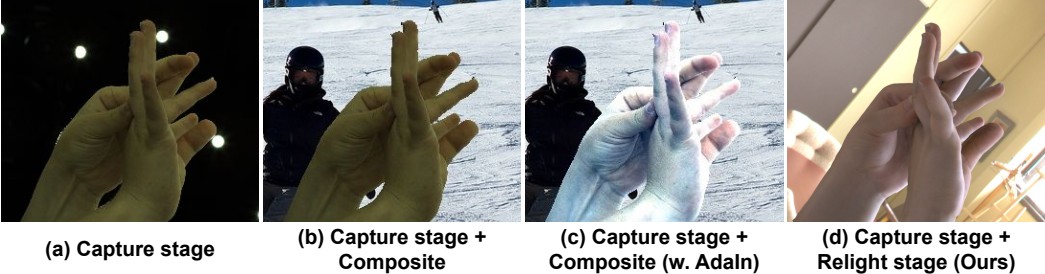

**(a) Capture stage**    **(b) Capture stage + Composite**    **(c) Capture stage + Composite (w. AdaIn)**    **(d) Capture stage + Relight stage (Ours)**

Figure 8: Image examples of datasets constructed in Tab. 2.

One advantage of our approach is that we can render images with any novel camera parameters. In addition to the above pre-defined 3rd-person viewpoints, we also render relighted images from random egocentric viewpoints to contribute our Re:InterHand to the egocentric 3D hand community. To this end, we first manually put a reference camera in the middle of two eyes using 3D scans that include both hands and a face. The orientation of the reference camera is set to see the center of the hands. Then, for each frame, we randomize 3D camera positions within 20 cm of a 3D box around the reference camera. We also randomize the 3D orientation of the camera by applying $[-30°, 30°]$ pitch, yaw, and roll. The principal point is set to the image center, and the focal length is randomized from $[0.7, 1.8]$ times the image size. To simulate the fisheye cameras, often used for egocentric viewpoints, we randomize distortion of the fisheye cameras by pre-defined mean and standard deviation. For the frame-based research, we render images with a different environment map for each frame at 30 fps, which results in 148K images. Also, for the video-based research, we render images with a different environment map for each segment at 30 fps, which results in 148K images.

For each peak pose sequence, we exclude frames at the first and last segment whose velocity of hands is less than a threshold to remove many neutral pose frames. Both 3rd-person and egocentric viewpoints images are rendered in 1K resolution.

**Non-binary masks.** We provide non-binary masks, obtained from the relight stage. The non-binary mask is different from binary masks rendered from MANO fittings as the non-binary ones are perfectly aligned with images including detailed silhouettes, such as nail and muscle bulging.

**3D hand model fittings.** We provide MANO [40] fitting as it is the most widely used 3D hand model in the community. Also, we provide the 3D hand model fittings, used to render relighted images.

## 5 Experiments

For all experiments, we report right hand-relative vertex error (RRVE), a Euclidean distance (mm) between estimated and GT 3D meshes of two hands after aligning translation of the right hand's root joint (*i.e.*, wrist). Note that the most widely used metric of previous works [58, 22, 26] (MPVPE) is calculated after aligning the translation of the right and left hand separately; hence, their MPVPE does not consider relative position between two hands, while our RRVE does. For the 3rd-person viewpoint experiments, we report RRVE on the test split of InterHand2.6M (H) [30], HIC [52], and the test split of our Re:InterHand. For the egocentric viewpoint experiments, we report RRVE on the test split of our Re:InterHand after training methods on the training split of it. For all experiments,

Table 3: Benchmark of state-of-the-art methods with the RRVE metric. Methods with † are tested with GT hand boxes. Methods with * indicate that they are trained additionally on Re:InterHand. We use the 3rd-person viewpoint split of our Re:InterHand.

| Methods | Testing sets | | |
|---|---|---|---|
| | InterHand2.6M | HIC | Re:InterHand |
| IntagHand [22]† | 19.96 | 67.11 | 52.91 |
| InterWild [26] | 19.74 | 23.59 | 37.59 |
| InterWild [26]* | **19.40** | **21.36** | **20.07** |

Table 4: Benchmark on the egocentric viewpoint split of our Re:InterHand.

| Methods | RRVE |
|---|---|
| InterWild [26] | 28.89 |

the frame-based split of Re:InterHand is used. For all datasets, the errors are calculated only for two-hand samples.

**Effectiveness of the relight stage.** Tab. 2 shows the effectiveness of the relight stage. It is noteworthy that our relight stage greatly reduces the error of HIC, which consists of real and natural images. Please note that images of HIC are entirely novel ones as they are only used for the testing purpose. Our relight stage also significantly reduces the test error on our Re:InterHand test set while slightly reducing errors on InterHand2.6M. As both InterHand2.6M and data from the capture stage consist of lab images, the data from the capture studio (the second row of the table) reduces the error on InterHand2.6M the most. However, it could not improve the test results on HIC with real and natural images as image appearances are far from those of real images, as shown in Fig. 8 (a). The variants with composition (the third and fourth rows) make the performance on HIC of the baseline (the first row) worse. We think the reason is that their images have inconsistent light between foreground and background, as shown in Fig. 8 (b) and (c). For more harmonic images, we applied AdaIn [16] to raw RGB pixels of the foreground to make them follow distributions of the background pixels. Unfortunately, as it is not aware of the reflections, it often changes the colors of hands to unrealistic colors without preserving skin colors and only changing lights, which results in performance degradation on HIC with real and natural image appearances.

**Benchmark.** Tab. 3 and 4 provide benchmark results with IntagHand [22] and InterWild [26], state-of-the-art 3D interacting hands recovery methods. We use their official checkpoints, and GT hand boxes are used for IntagHand as it assumes them.

## 6   Conclusion

**Summary.** We present Re:InterHand dataset, which provides images with highly realistic and diverse appearances of interacting hands and their corresponding GT 3D hands. To this end, our accurately tracked 3D poses, the state-of-the-art relighting network [17], and a number of high-resolution environment maps are used. We hope our dataset can make the community one step closer to the 3D interacting hands recovery in the wild.

**Limitations.** As Fig. 5 shows, our rendered images have cut at the forearm area. This is because our relighting network only takes a 3D hand geometry, not a whole-body one. We think this is not a severe issue as most 3D hand analysis systems take cropped hand images followed by hand detectors, where hand detectors can be trained on large-scale real datasets only with 2D annotations. Also, we observed that there are sometimes artifacts in the relighted images. This is because the relighting network is trained on single-hand data and tested on two-hand data, which sometimes results in pose generalization failure. We expect a better relighting network could alleviate this issue.

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
