# Supplementary Material of
# "A Dataset of Relighted 3D Interacting Hands"

**Gyeongsik Moon**
mks0601@meta.com

**Shunsuke Saito**
shunsukesaito@meta.com

**Weipeng Xu**
xuweipeng@meta.com

**Rohan Joshi**
rohanjoshi@meta.com

**Julia Buffalini**
jbuffalini@meta.com

**Harley Bellan**
harleybellan@meta.com

**Nicholas Rosen**
nicholasrosen@meta.com

**Jesse Richardson**
jesserichardson@meta.com

**Mallorie Mize**
malloriemize@meta.com

**Philippe de Bree**
phillippedebree@meta.com

**Tomas Simon**
tsimon@meta.com

**Bo Peng**
bopeng@meta.com

**Shubham Garg**
ssgarg@meta.com

**Kevyn McPhail**
kmcphail@meta.com

**Takaaki Shiratori**
tshiratori@meta.com

Meta Reality Labs Research

In this supplementary material, we provide more experiments, discussions, and other details that could not be included in the main text due to the lack of pages. The contents are summarized below:

1. Pose examples of Re:InterHand
2. Privacy

## A    Pose examples of Re:InterHand

Fig. A and B show pose examples of our Re:InterHand dataset.

## B    Privacy

All data captures are done after obtaining signatures from subjects with Meta's consent form. Our Re:InterHand dataset does not have personally identifiable information or offensive content.

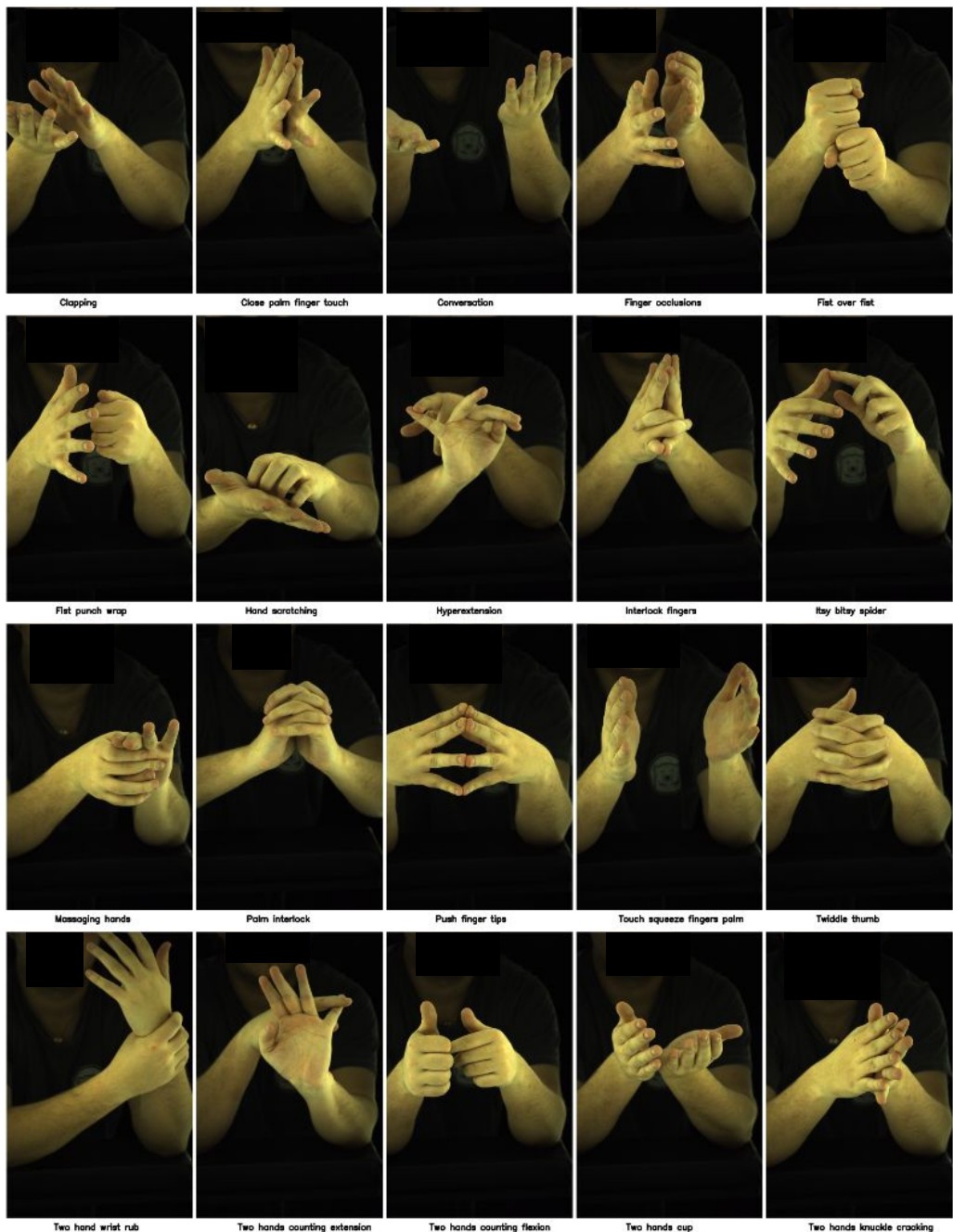

Figure A: Pose examples of our Re:InterHand dataset. Identifiable information is removed.

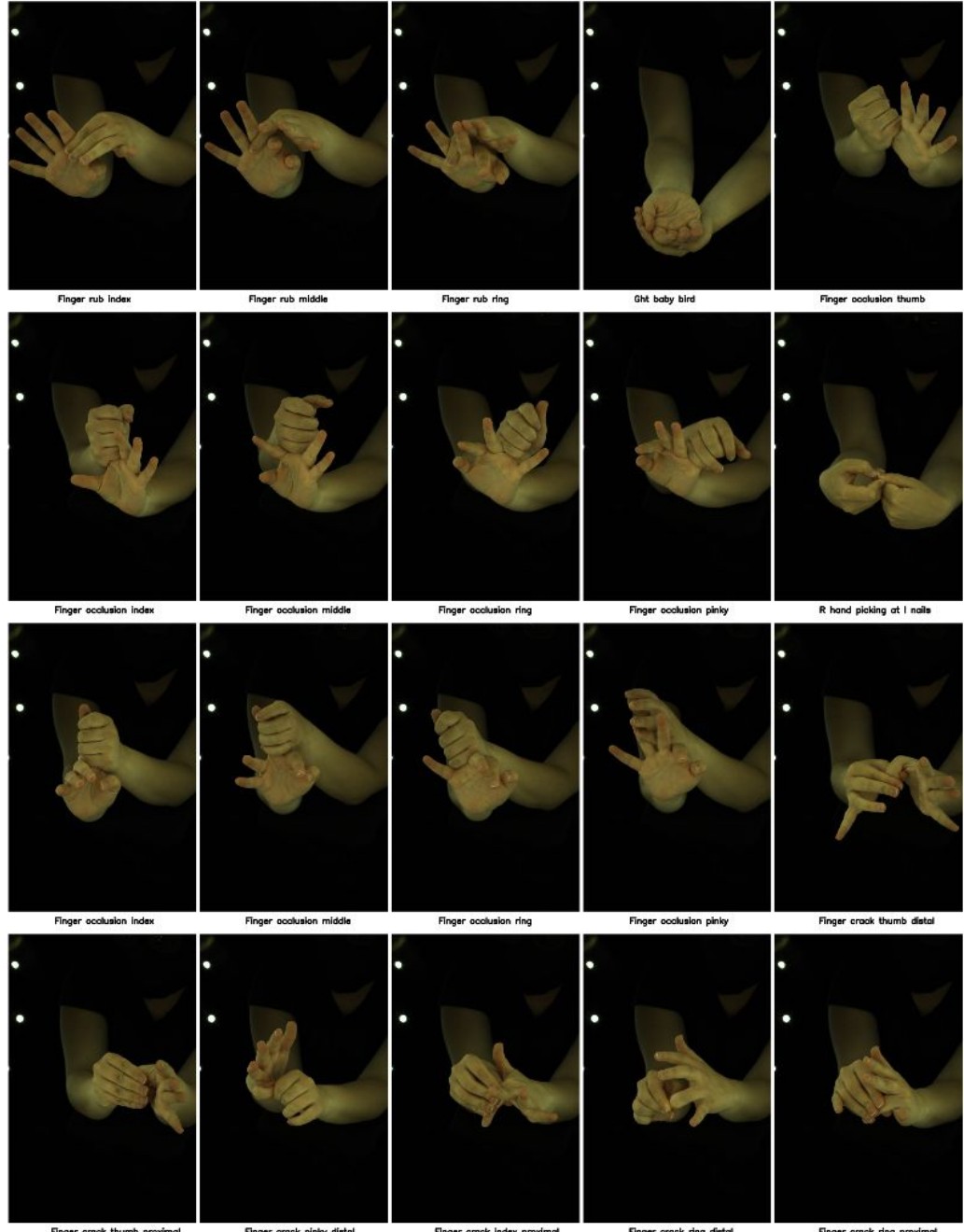

Figure B: Pose examples of our Re:InterHand dataset. Identifiable information is removed.