# OpenReview forum: "A Dataset of Relighted 3D Interacting Hands"
_NeurIPS.cc/2023/Track/Datasets_and_Benchmarks — NeurIPS 2023 Datasets and Benchmarks Poster_

### Official Review · Reviewer_vVQs · 2023-07-20
**Rejection**

**Rating:** 4
**Confidence:** 5
**Correctness:** The method proposed is sound.
**Clarity:** The English is clear enough to unders…

**Strengths:**

The proposed dataset is large in scale and have relatively good appearances.

**Additional Feedback:**

None

**Documentation:**

It is clearly stated.

**Limitations:**

The quality of dataset proposed is limited; while the dataset is large in scale.

**Opportunities For Improvement:**

As indicated in Fig.1, appearance is not realistic enough. It's only middle in the quality. Not close to the in-the-wild images.
Some points in Table 1 seems wrong: AssemblyHands and ARCTIC are having two-hand interactions; while authors express them as `x'.
Unless the appearance is fully realistic, I think one can trivially collect many samples with accurate ground-truths. Compared to AssemblyHands and ARCTIC, I cannot see much benefit for this paper.

**Relation To Prior Work:**

Some references are mis-leading in Table 1, as mentioned in weakness section.

**Summary And Contributions:**

The authors proposed the dataset for 3D interacting hands. The dataset is collected to achieve 1) diverse and realistic image appearances, 2) diverse and large-scale ground-truth 3D poses. Experiments demonstrate the benefits of the collected dataset compared to previous datasets.

---

> ### Author Response · Authors · 2023-08-04
>
> Dear Reviewer vVQs,
>
> Thanks for providing such a valuable suggestions and let us clarify reviewer's concerns.
> ARCTIC and AssemblyHands mainly consist of images where two hands are interacting with objects, not two hands are interacting with each other. Therefore, their 3D pose distribution is very different from ours, and we think the purpose of ARCTIC/AssemblyHands is different from 3D interacting hands datasets, such as HIC, InterHand2.6M and our Re:InterHand.
>
> 1. As Fig. 1 shows, our new dataset provides images with much more diverse appearances than Lab/Natural datasets, which include the suggested ARCTIC and AssemblyHands datasets. As ARCTIC is captured from a capture studio, its image appearance is restricted and far from those of in-the-wild images. AssemblyHands is captured only at a desk, which also makes its image appearances restricted. On the other hand, our images are rendered with diverse environmental maps including indoor and outdoor with various lightings. Table 2 shows that although our image appearances are not perfectly same as those of in-the-wild images, it is still useful to improve 3D interacting hands recovery on natural dataset (HIC).
>
> 2. Another advantage of our dataset is a free-viewpoint rendering. ARCTIC and AssemblyHands only provide images from fixed viewpoints; hence, models trained on their images would not generalizable to novel cameras with different extrinsic/intrinsic/distortion parameters. This becomes a serious issue for the egocentric viewpoints as many VR headsets and AR glasses use their own camera extrinsic/intrinsic/distortion parameters without sharing standard camera parameters. As we randomly perturb camera parameters for the rendering of the egocentric images (L219), regressors trained on our egocentric images could become robust to diverse egocentric cameras of VR headsets and AR glasses.
>
> 3. We're afraid Reviewer vVQs missed Fig. 2 of the main manuscript.
> * The left of Fig. 2 shows that ARCTIC has much longer distance between two hands on average (18.25 cm) compared to HIC (4.61 cm), InterHand2.6M (4.04 cm), and our Re:InterHand (3.84 cm). We additionally checked that AssemblyHands has 7.0 cm, much longer distance between two hands than HIC/InterHand2.6M/Re:InterHand.
> * The right of Fig. 2 shows the ratio of samples where two hands are contacting. Only very few samples in ARCTIC (2.37 %) have contacting two hands, while many samples in HIC (20.58 %), InterHand2.6M (44.74 %), and Re:InterHand (52.70 %) have contacting two hands. We additionally checked that AssemblyHands only has 13.38 %, much smaller than HIC/InterHand2.6M/Re:InterHand.
>
> The third point indicates that although ARCTIC and AssemblyHands have two-hand annotations, their two hands are far distant without many contact. Therefore, it is hard to say ARCTIC and AssemblyHands have interacting hand samples. Instead, they consist of images where two hands are interacting with objects, not two hands are interacting with each other, which makes they have totally different purpose than 3D interacting hand datasets, such as HIC/InterHand2.6M/Re:InterHand. As AssemblyHands does not provide 3D mesh annotations, we obtained them following InterHand2.6M to get those numbers.

---

> > ### Comment · Reviewer_vVQs · 2023-08-19
> > **Arguing for the rejection**
> >
> > Thanks for the clarification.
> >
> > While authors insisted that the environments of ARCTIC and AssemblyHands could be limited; however, I want to point out that they are still real images. We cannot say that it is 'realistic' for edited images; however the proposed data have edited images. We might need to validate its 'realism' seeing the difference in distributions between real and the proposed data; while there seems no such analysis in the draft. So, I think we cannot say that the appearance is 'realistic' yet. Also, seeing examples in Fig. 5, there is hands without elbows. This even lowers the quality of the data.
> >
> > Ignoring the first point, authors try to insist benefits compared to other benchmarks. My point is: it is hard to collect large-scale 'real' images with 'accurate' annotations. However, it is rather trivial to collect large-scale images with accurate annotations if we do not have to collect 'real' images. In my opinion, there could be other alternatives to this draft in such a case. It might be another issue to verify that this draft is the optimal compared to other such alternatives.
> >
> > Fig. 2 shows the benefits of the draft; while it seems rather incremental compared to InterHand2.6M in its characteristic. Also, Table 2 is not producing the SOTA results compared to [26]. This might be due to the difference in distribution. Even though it is in the challenging setting (not trained on HIC or InterHand2.6M); while if the dataset is truly realistic and in the large-scale, I think the accuracy needs to be close to SOTA.
> >
> > Last, I still think in Table 1, `Two-hand interactions' columns are mis-led. Even though it has larger distances between hands, several benchmarks such as H2O, ARCTIC and AssemblyHands are still having two hands and their interactions. They are different to single hand benchmark (such as FreiHand, STB, etc.).
> >
> > In Table 1, authors might want to insist that 'Image appearance' and 'Two-hand interactions' columns are distinct characteristics of the draft; while from above points, I am still questionable about this and thus, I am still leaning towards rejection.

---

> > > ### Author Response · Authors · 2023-08-19
> > >
> > > Dear reviewer vVQs,
> > >
> > > **About the realism of our dataset.**
> > > We think there is a big misunderstanding. We are not arguing that ours is the best in terms of all perspective than datasets with real images. Instead, we said that our dataset is to **complement existing datasets as all of each dataset has its own limitations (L50)**. Our dataset has not perfectly realistic images, which is the reason why we said 'middle' in Table 1. However, its image appearances are much more diverse than InterHand2.6M or other datasets captured from studios, such as ARCTIC and AssemblyHands. Table 2 shows that training InterWild on (InterHand2.6M+MSCOCO+dataset of the capture stage like Fig. 7 (a). second row of the table) gives worse results than InterWild trained on (InterHand2.6M+MSCOCO+our relighted dataset. last row of the table) on HIC dataset. Please note that HIC is captured in a daily environment (not from capture studio) and consist of real images. This shows that collecting more images from capture studio (like InterHand2.6M, ARCTIC, and AssemblyHands) are less helpful for improving 3D interacting hands regressor. Instead, our new dataset is necessary as it could **complement** existing datasets better. Please note that we already mentioned the forearm cut problem at L279.
> > >
> > > **Our dataset is easy to collect?**
> > > We think our dataset is not easy to collect and there are many research points to be considered. First, we need highly accurate and realistic 3D geometry of two hands. This is because we render images based on tracked 3D geometry of two hands to secure natural 3D hand motions. To this end, we use V2V-PoseNet to get much better 3D geometry than the triangulation of InterHand2.6M, as shown in Figure 6. Also, **Capture_compare_coord.mp4** and **Capture_compare_fit.mp4** of the supplementary material show the effectiveness of using the V2V-PoseNet compared to the triangulation of InterHand2.6M. In addition, we need state-of-the-art relighting network to render realistic images. On the other hand, existing methods, such as Ego3DHands and DART, simply changed backgrounds by alpha blending like Figure 7 (a) without considering light consistency between foreground and background. Table 2 shows that InterWild trained on (InterHand2.6M+MSCOCO+images from Figure 7 (b). third row of the table) or (InterHand2.6M+MSCOCO+images from Figure 7 (c). fourth row of the table) gives worse results than InterWild trained on (InterHand2.6M+MSCOCO+our relighted images. last row of the table). This shows that naively collecting synthesized datasets is less helpful than ours. Our dataset is collected with special considerations and not easy to collect.
> > >
> > >
> > > **Table 2 is not producing the SOTA results compared to [26]?**
> > > We think there is a big misunderstanding. **All results of Table 2 are from InterWild [26]. Please note that we used a better evaluation metric (L237) than previously used ones, the reason why numbers are different from previous ones. The variable of the table is the training set of InterWild.** We are greatly confused what reviewer wants to say by "Table 2 is not producing SOTA results compared to [26]". Please note that the first row of the table is evaluated with the official checkpoint of InterWild, trained on InterHand2.6M+MSCOCO and available from their website.
> > >
> > > **Table 1**
> > > Sorry for the misleading. We will change the column to (no, weak, and strong) instead of (check mark and x mark). H2O, ARCTIC, and AssemblyHands could be changed to weak interactions, while HIC, RGB2Hands, InterHand2.6M, Ego3DHands, and our Re:InterHand could be changed to strong interactions. Please note that we are addressing strong interactions as it is greatly challenging due to the severe occlusions between two hands, which barely exists in the datasets with weak interactions, such as H2O, ARCTIC, and AssemblyHands, as Figure 2 shows. We greatly thank reviewer for providing such a valuable suggestion and letting us clarify our paper more.

---

> > > > ### Comment · Reviewer_vVQs · 2023-08-19
> > > > **Comment by R#vVqs**
> > > >
> > > > Sincerely thanks for further clarification; while I still have a few concern.
> > > >
> > > > Authors said that they wrote 'middle' in Table 1, but it is not like that. In Fig. 1, authors denote the appearance as 'middle'; while in Table 1,  it is denoted as 'realistic'. I think this seems making confusion: it would be better to modify `realistic' in Table 1 to 'middle' as well.
> > > >
> > > > Then, authors proposed middle-quality; yet large-scale dataset that can be complementary to other benchmarks rather than proposing the best benchmark which is large-scale as well as in good quality. At this point, I am not sure if other reviewers clearly understand the situation, due to two mis-led points in Table 1.
> > > >
> > > > Besides, regarding this, my point is consistent: it is hard to collect large-scale 'real' images with 'accurate' annotations. However, it is rather trivial to collect large-scale images with accurate annotations if we do not have to collect 'real' images. I am questioning that such trivial tip is worth published in the top conference. I consistently argue that there could be other alternatives in such a case and this seems not the optimal.
> > > >
> > > > Regarding [26] issue, I expected that if the benchmark is worth to be published, it needs to be something which is truly large-scale and in good quality. Then, it can lead to the best performance in HIC. As far as I understand, the SOTA accuracy of HIC using InterWild is less than 20mm; while Table 2 is not producing better result than that. Authors argued that they can train better model combining their dataset gradually with others and experiments are showing that; while I do not think it is a sufficient way to show the effectiveness of the dataset which is qualified for the top venue.

---

> > > > > ### Author Response · Authors · 2023-08-19
> > > > >
> > > > > Dear reviewer,
> > > > >
> > > > > We think reviewer missed our updated rebuttal. Sorry for modifying them after posting it.
> > > > >
> > > > > **Table 1.** Although the appearance is middle-realistic, its diversity is much better than Lab images, such as InterHand2.6M. We tried to summarize all four properties (four rows of Figure 1) following Table 1 of AGORA paper. But we agree that we need more clarifications and will change the table following the reviewer's suggestion. Thanks for providing such a valuable suggestion.
> > > > >
> > > > > **Our dataset is easy to collect?**
> > > > > We think our dataset is not easy to collect and there are many research points to be considered. First, we need highly accurate and realistic 3D geometry of two hands. This is because we render images based on tracked 3D geometry of two hands to secure natural 3D hand motions. To this end, we use V2V-PoseNet to get much better 3D geometry than the triangulation of InterHand2.6M, as shown in Figure 6. Also, **Capture_compare_coord.mp4** and **Capture_compare_fit.mp4** of the supplementary material show the effectiveness of using the V2V-PoseNet compared to the triangulation of InterHand2.6M. In addition, we need state-of-the-art relighting network to render realistic images. On the other hand, existing methods, such as Ego3DHands and DART, simply changed backgrounds by alpha blending like Figure 7 (a) without considering light consistency between foreground and background. Table 2 shows that InterWild trained on (InterHand2.6M+MSCOCO+images from Figure 7 (b). third row of the table) or (InterHand2.6M+MSCOCO+images from Figure 7 (c). fourth row of the table) gives worse results than InterWild trained on (InterHand2.6M+MSCOCO+our relighted images. last row of the table). This shows that naively collecting synthesized datasets is less helpful than ours. Our dataset is collected with special considerations and not easy to collect.
> > > > >
> > > > > Compared to collecting datasets with real images with accurate 3D annotations, actually our dataset is a superset of them. The datasets with real images with accurate 3D annotation share the first challenge of above (accurate 3D annotation) with us. We addressed this by using V2V-PoseNet. In addition, we need to render realistic images after collecting real images with 3D annotations, which is an additional challenge of us, which does not exist for datasets with real images. Please note that the capture stage (Section 3.1) is a stage for capturing real images with accurate 3D annotations (where InterHand2.6M/ARCTIC/AssemblyHands belong to), and we have an additional relighting stage (Section 3.2). Therefore, we think ours needs much more considerations and not easy to collect.
> > > > >
> > > > >
> > > > > **Table 2 is not producing the SOTA results compared to [26]?** We think there is a big misunderstanding. **All results of Table 2 are from InterWild [26]. Please note that we used a better evaluation metric (L237) than previously used ones, the reason why numbers are different from previous ones. The variable of the table is the training set of InterWild.** Please note that the first row of the table is evaluated with the official checkpoint of InterWild, trained on InterHand2.6M+MSCOCO and available from their website.

---

> > > > > > ### Comment · Reviewer_vVQs · 2023-08-19
> > > > > > **Reviewer response**
> > > > > >
> > > > > > Thanks for the clarification.
> > > > > >
> > > > > > I missed the points authors mentioned regarding [26]. Now, I become more clear that authors got better accuracy compared to [26], which is very recent article (CVPR'23).
> > > > > >
> > > > > > I respect all the effort made by authors for collecting new dataset. I also agree that it was not an easy task. However 'easy to collect' might be different from 'trivial' that I mentioned. I am pointing out more about the novelty of the benchmark, while authors seemingly focused on the effort itself. At this point, I still can say that there could be other alternatives in such a case and even though it can marginally raise the accuracy; this seems not the optimal, especially due to edited images and forearm cut problem. I am still in this stance; while this seems rather subjective and I think other reviewers and chairs can fairly judge this issue.

---

> > > > > > > ### Author Response · Authors · 2023-08-19
> > > > > > >
> > > > > > > **1. Trivial to collect?**
> > > > > > >
> > > > > > > **We are greatly confused why the reviewer think our data collection is trivial compared to collecting datasets with real images and 3D annotations as our dataset is a superset of them**. Our dataset construction consists of two stages as shown in Figure 4.
> > > > > > >
> > > > > > > * First, the capture stage (Section 3.1) is a stage for capturing real images with accurate 3D annotations. The datasets with real images, such as InterHand2.6M/ARCTIC/AssemblyHands, belong to this stage.
> > > > > > > * Second, the relight stage (Section 3.2) is a stage for rendering realistic images with diverse appearances. **Please note that the second stage is performed based on assets from the first stage, not with some automatic and low-quality synthesizing tools. The relighting networks are also trained with assets from the first stage, not included in Figure 4.**
> > > > > > >
> > > > > > > As far as we understand, the reviewer is saying that 1) our dataset from the (capture stage + relight stage) is trivial to collect than 2) datasets only from the capture stage, such as InterHand2.6M/ARCTIC/AssemblyHands. We think this does not make sense as the capture stage is a subset of our two-stage pipeline. In other words, we have more things to consider than datasets only from the capture stage, such as InterHand2.6M. **All efforts and research problems of real image datasets, such as getting accurate 3D annotations from real images (Figure 6 and Capture_compare_coord.mp4 and Capture_compare_fit.mp4 of the supplementary material), are already addressed in our capture stage, and we additionally addressed realistic appearance problem in the relight stage.**
> > > > > > >
> > > > > > >
> > > > > > > **2. Not optimal? Maybe other alternative?**
> > > > > > >
> > > > > > > We would like to say that **all datasets (not only ours) have their own limitations as introduced in Section 1.** This means all dataset are not optimal except large-scale 3D interacting hands dataset with real and in-the-wild images, which is almost impossible to collect. **In this sense, recent 3D human community is focusing on utilizing synthesized datasets such as AGORA (CVPR 2021) and BEDLAM (CVPR 2023). In particular, AGORA whole-body split, which provide SMPL-X fits with body/face/hands poses, is still the most widely used one even it is published two years ago as obtaining 3D hands GT from in-the-wild images is almost impossible.** As an additional alternative, there have been several works that utilize other sensors such as IMU to get 3D GT of the 3D body from in-the-wild environment, such as 3DPW dataset. However, using other sensors for hands is not trivial. For example, due to the physical sizes and without clothes on hand, installing IMUs on hands should deteriorate hand image appearances. LiDAR/depth cameras are also not a possible option considering the small sizes of hands. In particular, the depth cameras do not work well in outdoor.
> > > > > > >
> > > > > > > Considering there have been no 3D interacting hands datasets with diverse and realistic image appearances, our dataset could serve an important position in 3D hands community as most of works mainly report their results on capture studio datasets, such as InterHand2.6M. **Robustness to in-the-wild images should be addressed for real-world applications, but lack of dataset makes it hard. We would like to contribute to this point with our dataset.** Also, we think it would be great if the reviewer could suggest some concrete candidates of better alternatives as saying "this work is not enough" without concrete examples makes hard to reach to a conclusion that all people agree. **We compared ours with possible alternatives in Table 2 with Figure 7, which include Lab dataset style and Composited dataset style of Figure 1.**
> > > > > > >
> > > > > > > **3. Novelty?**
> > > > > > >
> > > > > > > Our dataset has important novelty and contributions like below.
> > > > > > >
> > > > > > > * Ours is the first 3D interacting hands dataset that provide images with diverse and realistic appearances with accurate 3D annotations. Although the images are synthesized and have forearm cut problem, experimental results demonstrate usefulness of our dataset (**Table 2 and L279**).
> > > > > > > * We provide much better 3D annotations than existing the most widely used InterHand2.6M dataset by using learnable 3D keypoint annotator (**Figure 6 and  Capture_compare_coord.mp4 and Capture_compare_fit.mp4 of the supplementary material**). Also, we provide much more diverse 3D poses than InterHand2.6M (**Figure 3**).
> > > > > > > * As our dataset is a synthesized one, it supports free-viewpoint rendering. This is greatly helpful for egocentric case as there has been no 3D interacting hands dataset with egocentric images but we provide such egocentric images with simulated egocentric viewpoints. Also, as our simulated egocentric viewpoints are perturbed, our dataset could be used to various AR/VR headsets with diverse cameras extrinsics/intrinsics.
> > > > > > >
> > > > > > > Based on them, we think our dataset has considerable contributions. We hope the reviewer could recognize the significance of our work.

---

### Official Review · Reviewer_rLCe · 2023-07-21
**A realistic synthetic dataset for interacting hands.**

**Rating:** 10
**Confidence:** 4
**Correctness:** I think yes.
**Clarity:** The paper is well written.

**Strengths:**

[+] Contributing a large scale realistic dataset for interacting hands can benefit the future study for this area. (should be public released)

[+] Excellent analysis of previous works, highlighting their strengths and weaknesses.

[+] Providing comprehensive analysis for the dataset.

**Additional Feedback:**

Nope.

**Documentation:**

Yes.

**Ethics:**

Nope

**Limitations:**

Authors have adequately addressed the limitations and potential negative societal impact of their work.

**Opportunities For Improvement:**

I suggest to provide some error analysis in 2D/3D joint detection and 3D scan.

**Relation To Prior Work:**

Yes.

**Summary And Contributions:**

The paper proposes a pipeline to construct a large synthesis dataset for interacting hands. This pipeline aims to generate diverse hands' poses and realistic appearance. To achieve this, the paper:
1. captures multi-view images for two hands following InterHand.
2. obtains 3D hand mesh by combining 2D/3D joint detection, 3D scan, and 3D hand model fitting method, they
3. uses a SOTA relighting network to relight the 3D hand and produce realistic results.
The experiments show the proposed the dataset can help to improve the performance of 3D interacting hand reconstruction.

---

> ### Author Response · Authors · 2023-08-05
>
> Dear Reviewer rLCe,
>
> Thanks for recognizing a significance of our work.
> Let us answer questions of the reviewer.
>
> First, we checked that our 2D/3D keypoints are highly accurate.
> Our 2D keypoint detector has an error of 2.5 pixel in a 1024x667 image space.
> Also, our obtained 3D keypoint coordinates have an error of 2.0 mm.
> The errors are measured against our held-out human-annotated set.
> The errors are calculated without any translation/rotation/scale alignments.
>
> Unfortunately, there is no good way to check errors of 3D scans quantitatively because we do not have evaluation targets.
> For the 2D/3D keypoints, we evaluated against manually annotated ones; however, manually annotating 3D scans is impossible.
> Instead, we visually show our 3D scans of a capture in here: https://drive.google.com/file/d/1UJR6pTcqTqj9cuMJjeMTLqvk15tVtQ-q/view?usp=sharing
> Overall, our 3D scan has pixel-level accuracy and is temporally consistent.

---

> > ### Comment · Reviewer_rLCe · 2023-08-20
> > **keep the initial rate**
> >
> > Dear Authors,
> >
> > Thank you for your feedback on my questions. After reading the comments from other reviewers and authors, I have decided to maintain my initial rating.
> >
> > Firstly, I don't believe realism is a major issue when it comes to this topic. The synthesis of data holds significance across various subjects, including autonomous driving. As the difficulty in annotation, noisy label is a big problem in today's real-world data. A large synthesis data with high quality labels will benefit for pre-training models. The pre-trained models can be fine-tuned on a small set of high quality real-world dataset to achieve better real-world performance. So I think this dataset can benefit this topic a lot.
> >
> > Second, I satisfy with the experiments.

---

> > > ### Author Response · Authors · 2023-08-21
> > >
> > > Dear reviewer rLCe,
> > >
> > > Thanks for recognizing the significance of our work. As the reviewer said, as our relight stage renders images based on the 3D geometry, our 3D groundtruth is perfectly aligned with images.
> > >
> > > If the reviewer has more questions, please let us know and we are happy to answer them.

---

### Official Review · Reviewer_AoTd · 2023-07-21
**A Dataset of Relighted 3D Interacting Hands**

**Rating:** 7
**Confidence:** 3
**Clarity:** 1. Instead of directly mentioning tha…

**Strengths:**

The submission has several strengths:

1. The paper presents a clear and well-motivated introduction, setting the stage effectively for the research presented.

2. The introduction of the methodolgy in the paper is comprehensive, and the results demonstrate the benefits of the proposed dataset on 3D interacting hands recovery.

**Additional Feedback:**

Overall, the author's response to the concerns in previous sections is needed to make the final decision. I am happy to increase the rating if my concerns are addressed.

**Correctness:**

Overall, the claims in the submission is correct and the dataset is constructed in a sound way.

**Documentation:**

As stated, it lacks guidance on how to effectively utilize the data or provide a website for benchmarks and visualizations in the paper. Although the supplementary materials showcase promising results.

**Limitations:**

I think this work does not have negative societal impacts.

**Opportunities For Improvement:**

1. This work would benefit from additional detailed explanations in various places, given the availability of space in the paper. Please refer to the Clarity section.

2. Currently, the dataset is released; however, it lacks guidance on how to effectively utilize the data or provide a website for benchmarks and visualizations in the paper. Although the supplementary materials showcase promising results.

**Relation To Prior Work:**

Yes, the discussion of prior work is sufficient.

**Summary And Contributions:**

This paper introduces a novel and comprehensive large-scale dataset that addresses key limitations found in previous studies. The newly proposed dataset focuses on two-hand interactions, offering a more diverse background and simultaneously capture diverse 3D poses. The authors have made this dataset publicly available.

---

> ### Author Response · Authors · 2023-08-06
>
> Dear Reviewer AoTd,
>
> Thanks for recognizing a significance of our work.
> Let us answer reviewer's questions.
>
> 1. We are actually making our dataset bigger by rendering images from more subjects; hence, we are going to publicly release our dataset after the additional renderings. That is the reason why we haven't create a website yet. Before that, we'd like to show our prototype of the website, available in here (https://drive.google.com/file/d/117kVc2lxkL5YeWwesefSZbLs0sVRepLn/view?usp=sharing). Please open web.html with Google Chrome browser. After rendering from additional subjects, we will make the website include them and publicly open it. Please note that we will finish rendering soon and will make our dataset publicly available before NeurIPS conference starts to follow the guideline of NeurIPS Datasets and Benchmarks track. To ease the use of our dataset, we will also release dataloader and evaluator of our dataset, compatible with InterWild (https://github.com/facebookresearch/InterWild). Such dataloader and evaluator code could make people easily run a 3D interacting hands recovery system that uses our dataset both for the training and evaluation. Finally, we will release a pre-trained checkpoint of InterWild, trained additionally on our dataset.
>
> 2. Sorry for unclear descriptions in the manuscript. The peak pose is a sequence, which includes a transition from a neutral pose to a pre-defined pose (e.g., fist) and then transition back to the neutral pose. The purpose of the peak pose is to capture as diverse poses as possible including extreme poses and maximal finger bent. The range of motion is a sequence, which includes natural hand motion driven with minimal instructions, such as wave hands as if friends is coming over. In this way, we could capture both 1) diverse poses from the peak pose sequences and 2) natural hand motion from the range of motion sequences. We will clarify this in the camera-ready version.
>
> 3. As the caption of Fig. 6 says, the three frames has very short time difference from each other (0.02 seconds). Hence, the three frames should have almost the same 3D hands. The first row (a) not only suffers from the collisions, but also suffers from temporal inconsistency between very close frames. On the other hand, the second row (b) does not suffer from the collisions and achieves temporal consistency between close frames. We will clarify this in the camera-ready version.
>
> 4. Our two-hand tracking has high accuracy. Our 2D keypoint detector has an error of 2.5 pixel in a 1024x667 image space. Also, our obtained 3D keypoint coordinates have an error of 2.0 mm. The errors are measured against our held-out human-annotated set. Finally, we checked that the MANO meshes from NeuralAnnot have 1.3 mm errors from the 3D scans. The errors are calculated without any translation/rotation/scale alignments. Such accuracy is enough to render high-quality relighted images from trained relighting networks. Despite such a high accuracy, few frames could still fail due to the severe occlusions from the other hand. For such failed frames, we excluded them from the rendering after inspecting all frames manually (L193).

---

> > ### Comment · Reviewer_AoTd · 2023-08-21
> >
> > Thanks to the author for the rebuttal. My concerns have been addressed and I will update my rating. And glad to hear that the authors are working on expanding the dataset.

---

> > > ### Author Response · Authors · 2023-08-21
> > >
> > > Dear reviewer AoTd,
> > >
> > > Thanks for recognizing our rebuttal. If the reviewer has more questions, please let us know and we are happy to answer them.

---

### Author Response · Authors · 2023-08-11
**Reminder**

Dear reviewers,

We'd like to remind you that we posted an individual rebuttal to each reviewer.
Please feel free to ask any questions if you have after reading the rebuttal.

Best,
Authors.

---

### Decision · Program_Chairs · 2023-09-22

**Decision:**

Accept (Poster)

**Comment:**

This paper presents a pipeline for capturing, relighting, and re-rendering interacting hands, and utilizes this pipeline to introduce a new, large-scale, realistic synthetic dataset of interacting hands. In comparison to existing hand datasets, the proposed dataset offers close two-hand interactions, a broader diversity of poses, appearances, and illumination conditions, all at a larger scale and with reasonably high realism. Additionally, the paper demonstrates the advantages of using this dataset with a state-of-the-art hand recovery method.

Pros: The paper presents a clear motivation and a comprehensive introduction. The designed synthesis pipeline utilizes state-of-the-art techniques to re-render high-quality hand images. Consequently, the paper contributes a large-scale, high-quality interacting hands dataset with diverse poses, appearances, and illumination conditions, from which the community will benefit. Moreover, the results clearly demonstrate the advantages of this proposed dataset for 3D interacting hand recovery.

Cons: Although the dataset captures a wide diversity of appearances and poses in-the-wild setting and reasonably realistic, it is a synthetic dataset, and its quality is not as high as real images. Additionally, there is room for improvement in the explanation of the algorithms used in the pipeline and in the analysis to enhance the overall presentation.

Reviewers are in agreement that the paper is well-motivated and clear, with a well-designed pipeline and a comprehensive dataset that addresses a gap in large-scale interacting hands in real-world scenarios. Following the rebuttal, there remains a concern regarding the dataset's quality. Although the re-rendered images may not fully match the realism of real images, their quality is reasonably high, and the experiments clearly demonstrate the dataset's benefits for 3D hand recovery. This pipeline and dataset are expected to be valuable contributions to the field.